# Dye Adsorption Mechanism of Glass Fiber-Reinforced Plastic/Clay Ceramics and Influencing Factors

**DOI:** 10.3390/polym13183172

**Published:** 2021-09-18

**Authors:** Hiroyuki Kinoshita, Koya Sasaki, Kentaro Yasui, Yuko Miyakawa, Toshifumi Yuji, Naoaki Misawa, Narong Mungkung

**Affiliations:** 1Department of Engineering, University of Miyazaki, 1-1 Gakuen-Kibanadai-Nishi, Miyazaki 889-2192, Japan; 2Suzuki Motor Corporation, 300 Takatsuka-cho, Minami-ku, Hamamatsu City 432-8611, Shizuoka Prefecture, Japan; road15024@outlook.jp; 3National Institute of Technology, Kagoshima College, 1460-1 Hayatochoshinko, Kirishima City 899-5193, Kagoshima Prefecture, Japan; yasui@kagoshima-ct.ac.jp; 4Graduate School of Engineering, University of Miyazaki, 1-1 Gakuen-Kibanadai-Nishi, Miyazaki 889-2192, Japan; hj16051@student.miyazaki-u.ac.jp; 5Faculty of Education, University of Miyazaki, 1-1 Gakuen-Kibanadai-Nishi, Miyazaki 889-2192, Japan; yuji@cc.miyazaki-u.ac.jp; 6Center for Animal Disease Control, University of Miyazaki, 1-1 Gakuen-Kibanadai-Nishi, Miyazaki 889-2192, Japan; a0d901u@cc.miyazaki-u.ac.jp; 7Department of Electrical Technology Education, King Mongkut’s University of Technology Thonburi, Bangkok 10140, Thailand; narong_kmutt@hotmail.com

**Keywords:** waste GFRP, reuse, adsorbent, ceramics, dyeing wastewater, reduction firing

## Abstract

The effective reuse of waste glass fiber-reinforced plastic (GFRP) is desired. We previously produced porous ceramics by firing mixtures of crushed GFRP and clay in a reducing atmosphere and demonstrated their applicability as adsorbents for the removal of basic dyes from dyeing wastewater. However, the primary influencing factors and the dye adsorption mechanism have not been fully elucidated, and the adsorption of acidic and direct dyes has not been clarified. In this study, adsorption tests were conducted, and the effects of the firing atmosphere, specific surface area, type of dye, and individual components were comprehensively investigated. The results showed that reductively fired ceramics containing plastic carbide residue adsorbed basic dye very well but did not adsorb acidic dye well. The clay structure was the primary factor for the dye adsorption rather than the GFRP carbide. The mechanism for the basic dye adsorption appears to have been an increase in specific surface area due to the plastic carbide residue in the ceramic structure, which increased the ion exchange between the clay minerals and the dye. By adjusting the pH of the aqueous solution, the GFRP/clay ceramic also adsorbed considerable amounts of direct dye, so the mechanism was determined to be ion exchange with the calcium component of the glass fibers.

## 1. Introduction

Glass fiber-reinforced plastic (GFRP) is used in various products that require low weight and high strength, such as automobile parts, small ships, and wind turbine rotor blades. However, most waste GFRP is sent to landfills without being reused because it is difficult to recycle with existing technologies [1,2,3,4,5,6]. In previous work, we proposed mixing crushed waste GFRP with clay and then firing the resulting mixture to produce porous ceramics (i.e., GFRP/clay ceramics) [7,8,9] with novel material properties for environmentally friendly products [10,11,12]. Products made from waste are generally more expensive than products made from virgin materials because of the additional costs from waste collection and reforming; thus, ensuring their usefulness is critical to their commercial viability [13].

Firing a mixture of GFRP and clay powders in an oxidizing atmosphere produces a ceramic with high porosity in which the clay matrix is reinforced by glass fibers. This leads to high water permeability, so the ceramic can be applied as a filtering material for turbid water treatment [14] and water-permeable paving blocks [15]. In contrast, firing the mixture of GFRP and clay in a reducing atmosphere increases the specific surface area of the produced ceramic. This may be because plastic carbides remain in the ceramic structure, which includes a large number of nano-sized pores [16]. To exploit the high specific area of these ceramics and a high ion exchange function of clay [17,18,19,20,21], we investigated their potential application as an adsorbent material to remove pigment from dyeing wastewater [16]. Adsorption tests with methylene blue (MB) dye verified that reductively fired GFRP/clay ceramics possessed remarkably high adsorption.

However, it is not yet clear why the specific surface area of the ceramic increases when it contains the carbide residue of GFRP. The primary influencing factors and the mechanism of MB adsorption have not been fully elucidated, and the adsorption abilities of the ceramics for acidic and direct dyes [22,23,24] are unknown. To address these issues, we performed adsorption tests to clarify the mechanism for dye adsorption of GFRP/clay ceramics. The effects of the firing atmosphere, specific surface area, type of dye, and individual components of the GFRP/clay ceramics were comprehensively investigated.

## 2. Materials and Methods

We produced two types of GFRP/clay ceramics fired in oxidizing and reducing atmospheres to clarify the differences in the specific surface area and pore size distribution depending on the firing atmosphere. The effect of the GFRP carbide on the specific surface area of reductively fired GFRP/clay ceramics was investigated by firing GFRP pellets in a reducing atmosphere and observing the microstructure. Then, adsorption tests were performed with basic, acidic, and direct (azo) dyes to evaluate the dye adsorption abilities of the two types of GFRP/clay ceramics. Tests were also conducted with the glass fibers contained in the GFRP and GFRP carbide to identify the primary influencing factors for dye adsorption. In addition, the basic dye adsorption test on a ceramic containing plastic carbides but not glass fibers was conducted.

### 2.1. GFRP/Clay Ceramic Samples Used for Dye Adsorption Tests

Figure 1 shows the process used to prepare the GFRP/clay ceramic samples for the dye adsorption tests [16]. Polyamide (PA) thermoplastic pellets (Renny, Mitsubishi Engineering-Plastics Co., Tokyo, Japan) containing ~40% glass fiber by mass were used as a surrogate for waste GFRP. The clay (Sougoo Co., Miyakonojo, Japan) was produced in Miyazaki, Japan [25], and it is typically used in brick or tile manufacturing.

Table 1 presents the inorganic chemical compositions of the GFRP and clay after firing at 1073 K, which we measured using an energy-dispersive X-ray analyzer (EDX-720, Shimadzu Corporation, Kyoto, Japan) with the fundamental parameter method [16]. The GFRP contained E-type glass fibers, although the CaO content was high. The glass fibers had a diameter of ~10 µm and a length of ~1.0 mm. The major minerals of the clay are derived from the chlorite group.

Figure 2 shows examples of microscope (SZX10, Olympus Corporation, Tokyo, Japan) images of the samples [16]. The reductively fired samples were black because some of the undecomposed resin components remained in the clay structure as carbides.

Table 2 presents the sample manufacturing conditions [16]. The sample manufacturing procedure was as follows [16]:Clay was crushed with a rotary mill (New Power Mill ABS-W, Osaka Chemical Co., Ltd., Osaka, Japan) and then was sifted with a 0.3-mm mesh screen.GFRP was also crushed with the rotary mill and then was sifted with a 0.5-mm mesh screen.The crushed GFRP was mixed with the clay at the mass rates listed in Table 2.The GFRP/clay mixture was solidified by being pressed into a mold at 10 MPa. The molded samples had a diameter of 74 mm and a thickness of 50–60 mm.The molded samples were heated in an oxidizing or reducing atmosphere to 1073 K in an electric furnace (KY-4N, Kyoei Electric Kilns Co., Ltd., Tajimi, Japan). The samples were then held at the firing temperature for 1 h and allowed to cool to room temperature in the furnace. For oxidative firing, the samples were heated at 100 K h^−1^ to the firing temperature. For reductive firing, the samples were heated at 400 K h^−1^. The reducing atmosphere was obtained by closing the intake port attached to the bottom of the electric furnace.The produced GFRP/clay samples were then crushed with a hammer, and particle sizes of 0.5–1.0 mm were selected.

To clarify the primary influencing factors for dye adsorption of GFRP/clay ceramics, the glass fibers and GFRP carbides were prepared separately. The glass fibers were prepared by heating GFRP pellets at a rate of 100 K h^−1^ to 1073 K in an oxidizing atmosphere and then holding them at the firing temperature for 1 h to decompose the plastic component. GFRP carbides were prepared by heating GFRP pellets at a rate of 400 K h^−1^ to 1073 K in a reducing atmosphere. The samples were adjusted to a particle size of 0.5–1.0 mm by sieving after crushing.

In addition, we produced polyoxymethylene (POM)/clay ceramics containing plastic carbides but not glass fibers by firing a mixture of crushed POM resin (Iupital, Mitsubishi Engineering-Plastics Co., Tokyo, Japan) and clay in a reducing atmosphere. We then investigated the relationship between the quantity of plastic carbides (i.e., carbon content) and the specific surface area and MB adsorption ability of the ceramic. The POM resin was used as a substitute for PA plastic without glass fibers because the latter is not available from manufacturers. The mixing ratios of the POM resin to the total mass were 6%, 12%, and 18%. Mixing 40% GF/GFRP with clay at a ratio of 10% corresponded to a resin mixing ratio of 6%. The carbon content of the POM resin pellets used as the raw material was 41%.

Table 3 presents the inorganic chemical compositions of the samples [16]. The GFRP/clay ceramic samples had a greater CaO content than the clay ceramic, and the ratio of the CaO content to the total mass of the ceramic increased with the GFRP mixing ratio because of the increasing glass fiber content.

Table 4 presents the carbon content of each sample, which was measured using an elemental analyzer (CHNS/O Analyzer 2400, PerkinElmer Inc., Waltham, MA, USA) [16]. The carbon contents of the oxidatively fired ceramic samples were ~0.25%, and those of the reductively fired samples were 0.85–1.12%. The reductively fired ceramic samples contained more plastic carbides than the oxidatively fired ceramic samples. The 40% GF/GFRP pellets used as the raw material for the samples had a carbon content of 30%.

Table 5 presents the apparent porosity and specific surface area of each sample [16]. The apparent porosity of each ceramic was measured using a mercury porosimeter (Auto Pore IV 9500, Micromeritics Instrument Corporation, Norcross, GA, USA). The specific surface areas of the samples were measured using a high-precision gas/vapor adsorption measurement instrument (BELSORP-max, MicrotracBEL Corp., Osaka, Japan). The clay ceramic possessed an apparent porosity of ~32%, whereas the GFRP/clay ceramic samples possessed porosities of ~66% at most. The GFRP/clay ceramic samples possessed about twice the porosity of the clay ceramic because a large number of pores in the clay structure were created by the decomposition of the resin component due to firing. The porosities of the oxidatively and reductively fired GFRP/clay ceramic samples were almost equivalent. The oxidatively fired GFRP/clay ceramic samples had smaller specific surface areas than the clay ceramic, and the specific surface area decreased as the GFRP mixing ratio increased. In contrast, the reductively fired GFRP/clay ceramic samples had the same or larger specific surface areas than the clay ceramic.

Figure 3 shows the pore size distributions of the samples, which were measured using the same high-precision gas/vapor adsorption measurement instrument that was used to measure the specific surface area [16]. For the oxidatively fired GFRP/clay ceramic samples, nano-sized pores in the structure decreased as the GFRP mixing ratio was increased. Consequently, the specific surface areas of the samples were assumed to decrease with an increasing GFRP mixing ratio. The nano-sized pores were also assumed to disappear because of the sintering of the clay and glass fibers with an increasing GFRP mixing ratio.

The reductively fired GFRP/clay ceramic samples had more nano-sized pores relatively than the oxidatively fired GFRP/clay ceramic samples. Therefore, the reason why the reductively fired GFRP/clay ceramic samples possessed higher specific surface areas seems to be that the structure contained many nano-sized pores. It is believed that the GFRP carbide residue in the clay structure contributed to the increase in the specific surface area. To verify this, we measured the pore size distribution and the specific surface area of GFRP carbide.

Figure 4 shows the pore size distribution and specific surface area of the GFRP carbides, along with a photograph of the GFRP carbides. Although the GFRP carbides had no pores smaller than 10 nm, they had a large specific surface area. Therefore, the main reason for the reductively fired GFRP/clay ceramics with a greater specific surface area than the oxidatively fired GFRP/clay ceramics was concluded to be because the former contained GFRP carbides. In addition, the clay matrix may have contained more fine pores because of the inclusion of plastic carbides.

Figure 5 shows examples of scanning electron microscope (SEM, S5500, Hitachi High-Technologies Corporation, Tokyo, Japan) images of the surface structures of the clay and 20% GFRP/clay ceramic samples. Compared with the oxidatively fired ceramics, the reductively fired ceramics tended to have smaller clay-sintered particles, although the difference is not clear. In addition, a large number of ultrafine particles were attached to the clay-sintered particles. However, it is unclear whether these ultrafine particles are plastic carbides.

### 2.2. Methodology of Dye Adsorption Tests

Figure 6 shows a schematic diagram of the dye adsorption test [16]. MB (C_16_H_18_N_3_SCl) (Fujifilm Wako Pure Chemical Corporation, Osaka, Japan), Orange II (HOC_10_H_6_N:NC_6_H_4_SO_3_Na), and Congo-red (C_32_H_22_N_6_Na_2_O_6_S_2_) were used as representative basic, acidic, and direct (azo) dyes, respectively. The test procedure was as follows [16]:Samples were washed with distilled water and were dried in an electric furnace at 373 K for over 24 h before the dye adsorption tests.MB, Orange II, and Congo-red powders were dissolved in distilled water to make aqueous solutions each with a concentration of 1 × 10^−4^ mol/L.Then, 1 g of the granular samples was placed in a beaker containing 50 mL of the aqueous solution, and the aqueous solution was stirred with a stirring device (EYLA ZZ-1010, Rikakikai Co., Ltd., Tokyo, Japan) at a speed of 150 rpm.The dye concentration and pH value of the aqueous solutions were measured after 1, 10, 30, 60, and 120 min.The dye concentration in the aqueous solution was measured using a drainage analyzer (NDR-2000, Nippon Denshoku Industries Co., Ltd., Tokyo, Japan). The absorbance of the aqueous solution was determined. Then, the dye concentration was calculated from a calibration curve that expressed the relationship between the absorbance and dye concentration of the aqueous solution. The pH of the aqueous solution was measured using a pH meter (HM-25R, DKK-TOA Corporation, Tokyo, Japan).

## 3. Results

### 3.1. Dye Adsorption of GFRP/Clay Ceramics

Figure 7 shows the dye concentration reduction rates of the oxidatively and reductively fired ceramic samples. For MB, all samples demonstrated considerable dye adsorption. However, the oxidatively fired GFRP/clay ceramic samples had a lower reduction rate than the clay ceramic sample, and the rate decreased as the GFRP mixing ratio increased. In contrast, the reductively fired GFRP/clay ceramic samples had a higher reduction rate than the clay ceramic sample, and the rate increased with the GFRP mixing ratio. For Orange II, the clay and oxidatively fired GFRP/clay ceramic samples did not adsorb the dye at all. In contrast, the reductively fired GFRP/clay ceramic samples adsorbed it to some degree. For Congo-red, all samples adsorbed the dye to a certain degree, and the oxidatively fired GFRP/clay ceramic samples had reduction rates comparable or slightly higher to those of the reductively fired samples. The results confirmed that the reductively fired GFRP/clay ceramics had a high adsorption capacity for MB, which is a basic dye.

Figure 8 shows the temporal change in pH of each dye solution during the dye adsorption test. The MB solution containing the clay ceramic sample was weakly acidic and eventually became almost neutral. In contrast, the MB solution containing the oxidatively fired GFRP/clay ceramic sample was alkaline, and the pH value increased with the GFRP mixing ratio because of the glass fibers, which mainly comprised calcium. The MB solution containing the reductively fired GFRP/clay ceramic sample was also alkaline. However, the pH value was slightly lower than that of the solution containing the oxidatively fired sample because it contained plastic carbides. The changes in pH of the solution also differed with the samples. The pH value of the solution containing the oxidatively fired GFRP/clay ceramic sample was almost constant over time, while that of the solution containing the reductively fired sample decreased gradually over time. This indicates that the concentration of hydroxide ions in the solution decreased gradually because of ion exchange. This neutralized the cations of MB in the solution, which caused the color of the solution to turn from blue to colorless.

The Orange II solution containing the clay ceramic sample was almost neutral, and the pH value was constant over time. The change in pH value was similar to that observed for the MB solution. In particular, the pH value of the solution containing the reductively fired GFRP/clay ceramic sample decreased over time. The Congo-red solution with the clay ceramic sample was weakly alkaline, and it approached neutrality over time. The change in pH value was similar to that observed for the MB and Orange II solutions. For all dye solutions containing the reductively fired GFRP/clay ceramic sample, the difference in pH value according to the GFRP mixing ratio was very small.

### 3.2. Primary Influencing Factors for Dye Adsorption on the GFRP/Clay Ceramics

To clarify the influencing factors for the dye adsorption of the GFRP/clay ceramics and the absorption mechanism, we conducted dye adsorption tests on the glass fibers in GFRP and on GFRP carbides. Figure 9 shows the dye concentration reduction rates. Figure 10 shows the temporal changes in pH of the dye solutions. The primary influencing factors and adsorption mechanism of each dye are discussed below individually.

(a)MB

Figure 9a shows that the glass fibers did not adsorb MB at all. The MB concentration reduction rate was lower with the GFRP carbides than with the clay ceramic sample. This indicates that the plastic carbides do not have a significant MB adsorption capacity on their own. Therefore, the primary factor for MB adsorption of reductively fired GFRP/clay ceramics must be the clay structure. This is also consistent with the fact that the MB concentration reduction rate for the oxidatively fired GFRP/clay ceramic samples decreased with an increasing GFRP mixing ratio. In particular, the reduction rate decreased as the quantity of clay was decreased.

(b) Orange II

Figure 9b shows that the clay ceramic sample did not adsorb Orange II at all, and the glass fibers and GFRP carbide adsorbed a small amount. Therefore, the primary factor for dye adsorption of the GFRP/clay ceramics is the glass fibers, and the plastic carbide offers no adsorption of acidic dyes.

(c) Congo-red

Figure 9c shows that the GFRP carbides did not adsorb Congo-red at all, while the glass fibers showed considerable adsorption. Therefore, the primary factor for dye adsorption of the GFRP/clay ceramics is the glass fibers. The plastic carbide did not adsorb the dye at all and was actually a hindrance. This explains why the dye concentration reduction rate was slightly higher for the oxidatively fired GFRP/clay ceramic without plastic carbides than for the reductively fired ceramic, as shown in Figure 7.

## 4. Discussion

The results of dye absorption tests showed that while reductively fired GFRP/clay ceramics adsorbed MB dye very well, they did not adsorb very much Orange II and Congo-red dyes. We first discuss the reason for this. In this study, the chlorite clay used as a base material for the GFRP/clay ceramics had a predominant permanent charge and a cation exchangeable capacity of 5–40 [17]. In addition, when the aqueous solution containing a ceramic sample is alkaline, as was the pH of the aqueous solution in this study, the surface of the ceramic sample is negatively charged [17,26,27]. Therefore, ceramics containing chlorite clay adsorb cations from the aqueous solution by ion exchange. For this reason, the MB dye, which existed as cations in an aqueous solution, was adsorbed onto the clay surface [20]. Moreover, the adsorption of basic dyes is caused by van der Waals forces in addition to cation exchange [17].

On the contrary, the acid anionic Orange II dye contained sulfonic acid in an aqueous solution. Therefore, when the surface of a ceramic is negatively charged, the dye will not be electrically adsorbed onto the ceramic surface. Congo-red, a direct dye, is also an anionic dye that contains sulfonic acid; however, the adsorption of Congo-red onto cotton and silk surfaces is mainly due to hydrogen bonding rather than ion exchange.

Second, we discuss why the glass fibers adsorbed the Congo-red dye well. Calcium-rich fly ash and cement adsorb Congo-red dye regardless of whether the aqueous solution is acidic or alkaline [28,29]. However, when the aqueous solution is alkaline, the adsorption amount decreases as the pH increases. The main cause of this dye adsorption is thought to be ion exchange with calcium ions [28]. Therefore, it is believed that the cause of the glass fibers adsorbing the Congo-red dye well in this study was ion exchange with calcium ions.

In addition, because GFRP/clay ceramics contain glass fibers, they have higher CaO contents compared to clay-only ceramics. When a Congo-red dye solution is alkaline, the adsorption amount is low, but when the solution is acidic, a larger amount can be expected to be adsorbed, which can be achieved by adjusting the pH of the aqueous solution [30,31].

Figure 11 shows the reduction rates of Congo-red dye concentrations and the change in pH of the dye solution when the initial pH of the dye solution containing the reductively fired 20% GFRP/clay ceramic sample was adjusted in the range of 3–8 by adding hydrochloric acid. The GFRP/clay ceramic adsorbed a considerable amount of Congo-red dye when the pH of the aqueous solution was 4–5.

However, in this study, the dye concentration of the aqueous solution was indirectly estimated by measuring its absorbance. The color of Congo-red solution changes from red to dark blue when the pH is approximately three or less [28]. Namely, the color of the aqueous solution becomes darker. Therefore, when the pH of the aqueous solution is adjusted to three or less, the relationship between the absorbance and the dye concentration should be recalibrated, but this is not easy and is under consideration. For this reason, the dye concentrations reduction rate at pH 3 shown in Figure 11a may contain relatively large measurement errors. It is necessary to perform Fourier-transform infrared measurements for more accurate examination. Moreover, the effect of carbides in the reductively fired GFRP/clay ceramic on the adsorption of Congo-red dye is still unknown. These issues should be addressed in the future.

Azo dyes are currently the most used dyes in the industry. Therefore, the development of an effective technique for removing the dye from the dyeing wastewater is strongly desired. Roberto et al. reported that Azo Dye Reactive Violet 5 (RV5) could be effectively removed using photocatalysis technology [32], and Zuorro et al. also showed the usefulness of RV5 dye removal technology using non-living cells of Nannochloropsis oceanica [33]. Further elucidation of the mechanism of azo dye adsorption and the development of its dye removal technology is desired.

Next, we discuss why the reductively fired GFRP/clay ceramics had greater MB adsorption than the oxidatively fired GFRP/clay ceramics. We attributed to the former containing plastic carbides and thus possessing a greater specific surface area. However, it is believed that the primary factor for MB adsorption of the ceramics is the clay structure because the plastic carbides did not have significant MB adsorption on their own. Therefore, we investigated the effect of increasing the specific surface area on the MB adsorption of the clay ceramic.

Figure 12 compares the MB concentration reduction rates of clay ceramics with different specific surface areas (a) and the pore size distribution of clay sample B, which was newly prepared for the dye adsorption experiment (b). Clay sample A was the clay ceramic sample used in the previous dye adsorption experiments, as listed in Table 2. This sample was produced by being heated at 100 K h^−1^ to the firing temperature (1073 K) in an oxidizing atmosphere. It possessed a specific surface area of 11.0 m^2^/g, as given in Table 5. In contrast, clay sample B was heated at 400 K h^−1^ to the firing temperature in a reducing atmosphere, similar to the reductively fired GFRP/clay ceramic samples. Clay sample B had fewer nano-sized pores than clay sample A (see Figure 3) and possessed a smaller specific surface area. The apparent porosity of clay sample B (31.6%) was comparable to that of clay sample A.

Figure 12a shows that clay sample B had a considerably lower MB concentration reduction rate. This result indicates that the MB adsorption of the clay ceramic strongly depends on the specific surface area. These results showed that increasing the specific surface area of the clay structure has a large effect on the MB adsorption of GFRP/clay ceramics. However, it is currently unknown why clay sample B had a smaller specific surface area than clay sample A. Further research is needed on the relationships of the heating rate and firing atmosphere with the specific surface area of the clay ceramic. This is also an issue to be addressed in the future.

In addition, we investigated whether the specific surface area of the ceramic increases when only plastic carbides remained in the clay structure and the MB adsorption ability of the ceramic containing plastic carbides but not glass fibers.

Table 6 presents the apparent porosities, specific surface areas, and carbon contents of the POM/clay ceramics containing plastic carbides but not glass fibers. The specific surface areas of the ceramics increased with the inclusion of the plastic carbides, although the specific surface area did not increase in proportion with the carbon content.

Figure 13 shows the MB concentration reduction rates of the ceramics. The MB adsorption of the ceramics increased with the porosity, as well as the specific surface area. For a more detailed investigation, measuring the specific surface area of the plastic carbide itself would be desirable. However, when the plastic (POM resin) without glass fibers was fired in a reducing atmosphere in the same manner as GFRP, almost no carbide remained. Therefore, it was impossible to measure the specific surface area of the plastic carbides.

The above results indicate that the MB absorption mechanism is as follows:The plastic carbide residue in the reductively fired GFRP/clay ceramic structure increases the specific surface area of the ceramic. The high specific surface area of the ceramic increases the physical and chemical adsorption of MB. In addition, the high porosity facilitates MB movement into the ceramic body.The reductively fired GFRP/clay ceramic samples exhibited a high adsorption capacity for only basic dyes. This indicates that the adsorption mechanism is dominated by cation exchange with clay components.

## 5. Conclusions

To realize the effective reuse of waste GFRP, we produced porous ceramics by mixing crushed GFRP with clay and then firing the resulting mixture in either an oxidizing or reducing atmosphere. We previously showed that these porous ceramics can be used as adsorbents to remove basic dye from dyeing wastewater. In this study, we conducted adsorption tests with MB, Orange II, and Congo-red dyes to evaluate the effectiveness of these ceramics with different types of dye. We also tested the glass fibers and GFRP carbides to clarify the primary influencing factors and mechanism of the dye adsorption. The results confirmed that reductively fired GFRP/clay ceramics containing plastic carbide residue in the ceramic structure adsorb MB very well. The clay structure was the primary factor for the dye adsorption rather than the GFRP carbide. The MB adsorption mechanism was based on the plastic carbide residue in the ceramic structure increasing the specific surface area, which increased ion exchange between the clay minerals and the dye. The GFRP/clay ceramic also adsorbed considerably more Congo-red dye when the pH of the aqueous solution was adjusted. The adsorption mechanism was based on ion exchange with the calcium component of the glass fibers.

Finally, although chlorite clay was used as the base material for the GFRP/clay ceramics in this study, the type of clay can be changed according to the type of dye. For example, the use of anionic clay minerals for acid dyes.

Adsorbents require both large pore volume and high specific surface area [34]; however, it is not easy to obtain a sintered product with a high specific surface area. The results of this study showed that plastic carbide residue increases the specific surface area of ceramics, so it would be of great significance if a method for increasing the specific surface area of sintered products was established. We also hope that the results of this study will contribute to the effective reuse of waste GFRP.

## 6. Patents

Kinoshita H, Kaizu K., Ikeda K., (2013) Manufacturing method of porous ceramics using waste GFRP, Japanese Patent No. 5167520 (in Japanese).

## Figures and Tables

**Figure 1 polymers-13-03172-f001:**
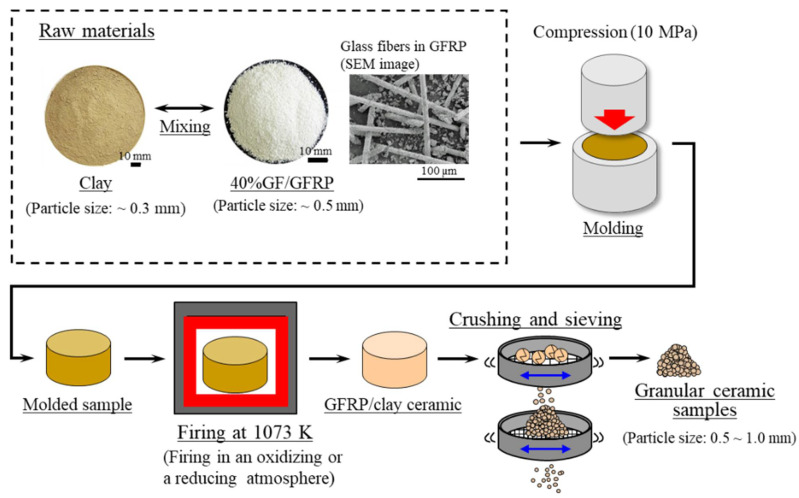
Manufacturing process of GFRP/clay ceramic samples used for the dye adsorption tests.

**Figure 2 polymers-13-03172-f002:**
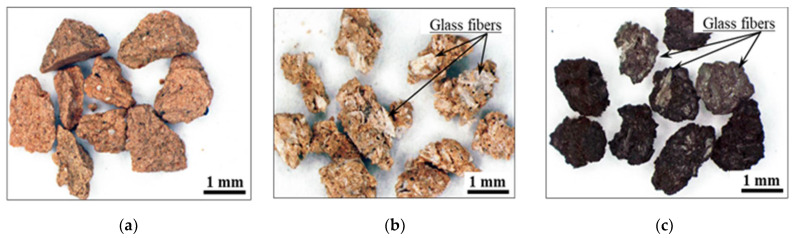
Microscope images of the samples. (**a**) Clay (**b**) 60% GFRP/clay (Oxidatively fired) (**c**) 60% GFRP/clay (Reductively fired).

**Figure 3 polymers-13-03172-f003:**
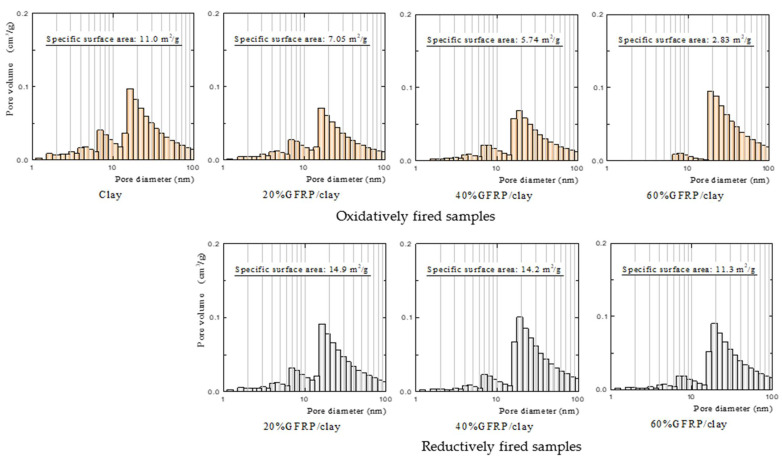
Pore size distributions of clay and GFRP/clay ceramic samples.

**Figure 4 polymers-13-03172-f004:**
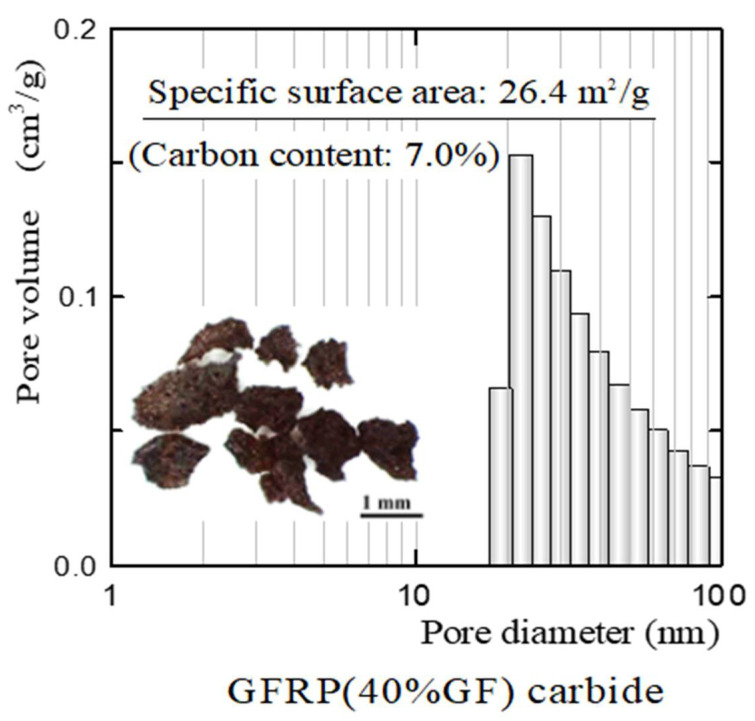
Pore size distribution and specific surface area of GFRP carbides.

**Figure 5 polymers-13-03172-f005:**
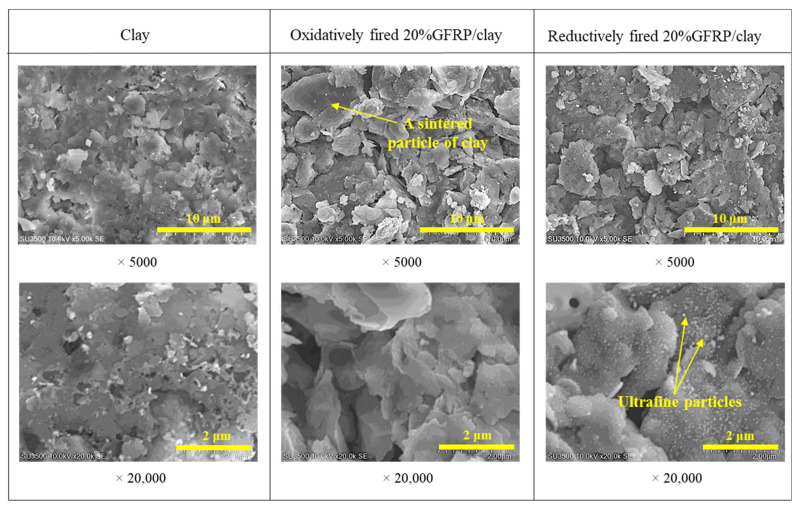
SEM images of the surface structures of clay and GFRP/clay ceramic samples.

**Figure 6 polymers-13-03172-f006:**
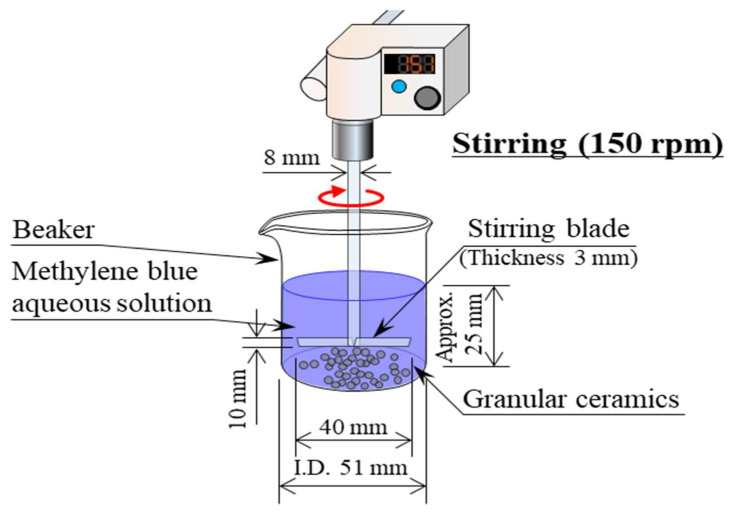
Schematic diagram of the dye adsorption test.

**Figure 7 polymers-13-03172-f007:**
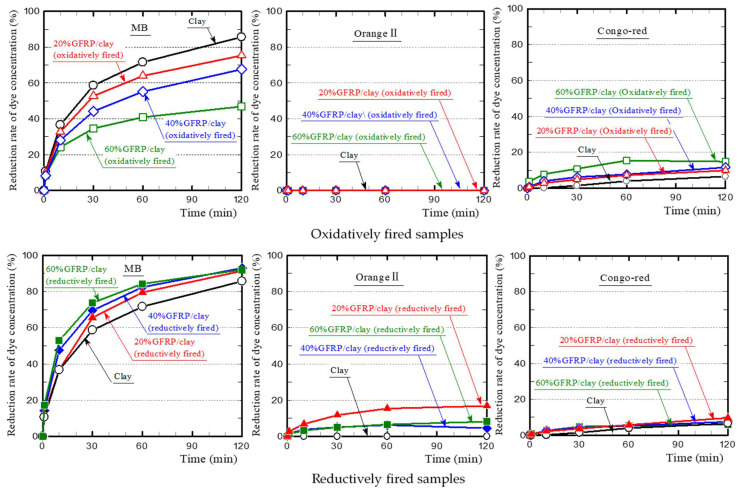
Dye concentration reduction rates of samples.

**Figure 8 polymers-13-03172-f008:**
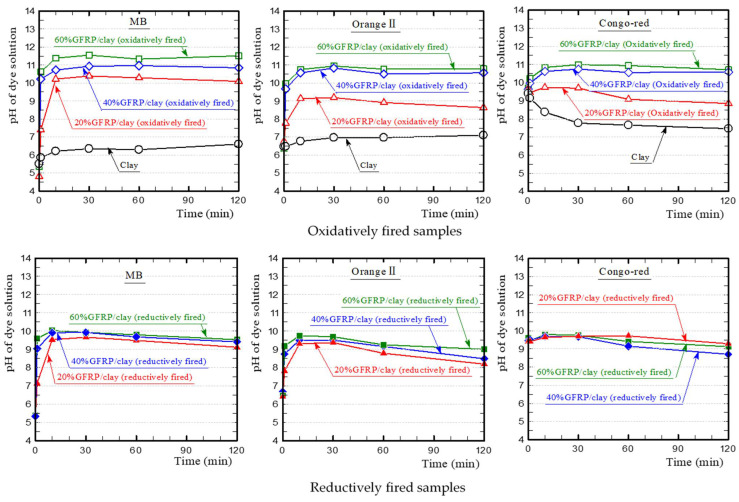
The temporal changes in pH of the dye solutions.

**Figure 9 polymers-13-03172-f009:**
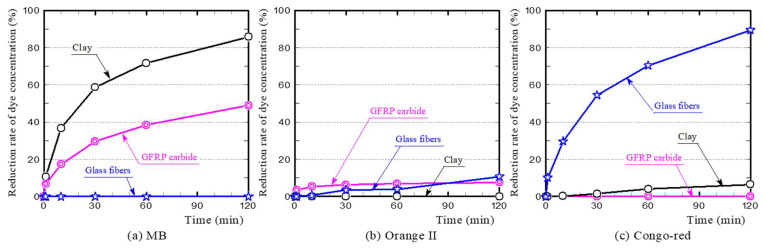
MB **(a)**, Orange II (**b**), Congo-red (**c**) dye concentration reduction rates of the GFRP carbides and glass fibers.

**Figure 10 polymers-13-03172-f010:**
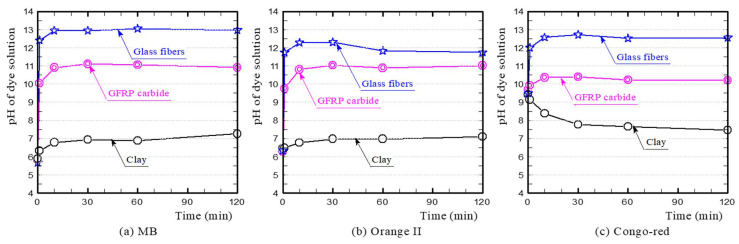
Temporal changes in pH of MB (**a**), Orange II (**b**) and Congo-red (**c**) dye solutions.

**Figure 11 polymers-13-03172-f011:**
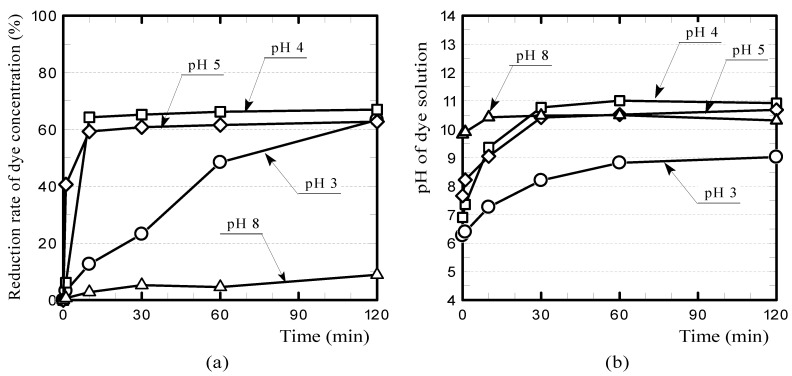
(**a**) Congo-red dye concentration reduction rates and (**b**) the change in pH of the dye Scheme 20. GFRP/clay ceramic sample was adjusted in the range of 3–8 by adding hydrochloric acid.

**Figure 12 polymers-13-03172-f012:**
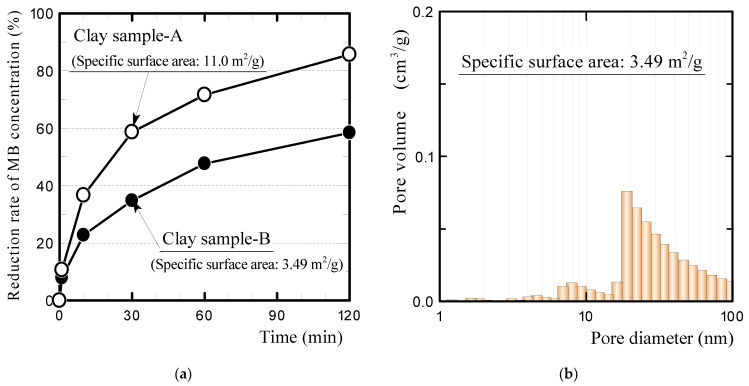
(**a**) Comparison of the MB concentration reduction rates for clay ceramics with different specific surface areas and (**b**) pore size distribution of clay sample B.

**Figure 13 polymers-13-03172-f013:**
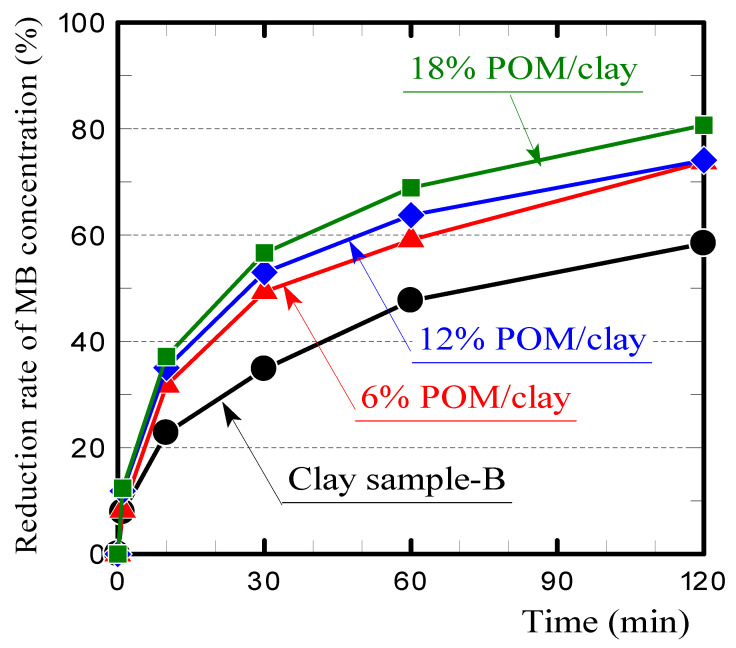
MB concentration reduction rates on POM/clay ceramics.

**Table 1 polymers-13-03172-t001:** Compositions of inorganic substances in the clay and GFRP.

Component	Raw Materials	
	Clay (Mass%)	40% GF/GFRP (Mass%)
SiO_2_	65.8	54.9
Al_2_O_3_	21.9	16.3
Fe_2_O_3_	4.79	0.77
K_2_O	3.37	0.15
MgO	1.67	-
CaO	1.31	26.7
TiO_2_	0.87	0.56
others	0.29	0.62

**Table 2 polymers-13-03172-t002:** Manufacturing conditions for GFRP/clay ceramic samples.

Samples		Mixing Ratios of GFRP (Mass %)	Firing Conditions
Oxidatively fired ceramics	Clay	0	Samples were heated at 100 K h^−1^ to 1073 K in an oxidizing atmosphere and then held at the firing temperature for 1 h
	20% GFRP/clay	20	
	40% GFRP/clay	40	
	60% GFRP/clay	60	
Reductively fired ceramics	20% GFRP/clay	20	Samples were heated at 400 K h^−1^ to 1073 K in a reducing atmosphere and then held at the firing temperature for 1 h
	40% GFRP/clay	40	
	60% GFRP/clay	60	

**Table 3 polymers-13-03172-t003:** Inorganic chemical compositions of the ceramic samples.

Component	Oxidatively Fired			Reductively Fired		
	20% GFRP/Clay	40% GFRP/Clay	60% GFRP/Clay	20% GFRP/Clay	40% GFRP/Clay	60% GFRP/Clay
SiO_2_	62.6	59.1	50.0	62.2	61.2	56.2
Al_2_O_3_	22.1	20.7	17.7	18.5	9.13	4.79
Fe_2_O_3_	4.87	4.16	4.09	6.13	7.56	7.34
K_2_O	3.26	2.91	2.00	3.73	3.77	3.11
MgO	1.66	1.75	1.51	2.24	2.43	2.14
CaO	4.02	9.93	23.2	5.34	12.9	22.7
TiO_2_	0.86	0.80	1.03	1.21	1.56	1.49
Others	0.58	0.71	0.45	0.65	1.46	2.22

**Table 4 polymers-13-03172-t004:** Carbon contents of samples.

Samples		Carbon Content (%)
Oxidatively fired ceramics	Clay	0.06
	20% GFRP/clay	0.24
	40% GFRP/clay	0.25
	60% GFRP/clay	0.26
Reductively fired ceramics	20% GFRP/clay	0.85
	40% GFRP/clay	0.99
	60% GFRP/clay	1.12

**Table 5 polymers-13-03172-t005:** Apparent porosity, specific surface area, and carbon content of each ceramic sample.

Samples		Apparent Porosity (%)	Specific Surface Area (m^2^/g)
Oxidatively fired ceramics	Clay	31.9	11.0
	20% GFRP/clay	38.2	7.05
	40% GFRP/clay	52.7	5.74
	60% GFRP/clay	62.9	2.83
Reductively fired ceramics	20% GFRP/clay	43.1	14.9
	40% GFRP/clay	53.8	14.2
	60% GFRP/clay	66.2	11.3

**Table 6 polymers-13-03172-t006:** Apparent porosity, specific surface area, and carbon content of POM/clay ceramic.

Samples	Apparent Porosity (%)	Specific Surface Area (m^2^/g)	Carbon Content (%)
Clay sample-B	31.6	3.49	0.27
6% POM/clay	29.5	7.32	0.46
12% POM/clay	33.2	7.48	0.58
18% POM/clay	46.7	7.19	0.58

## Data Availability

The data presented in this study are available on request from the corresponding author.

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
