# Peer review of "Dye Adsorption Mechanism of Glass Fiber-Reinforced Plastic/Clay Ceramics and Influencing Factors"

_polymers, 2021, doi:10.3390/polym13183172_

Round 1
Reviewer 1 Report
The manuscript described the preparation of porous ceramics based on glass fiber-reinforced plastic/clay composites, followed by their use in dye removal from aqueous solutions. Although the topic itself has few adherence to a polymer journal, the text is mostly clear and concise. I support its publication after the below listed changes.
Written elements in Figure 3 are unreadable. Please, correct that;
Experiments of dye adsorption at different pH values should be better addressed. They are not refereed in the experimental part, but appear in Figure 11 (discussion) only for congo red and in a short range of pH (3-5). Did the authors try to vary this parameter for other dyes and different pH ranges?
In the same line with the last comment, zeta potential measurements on adsorbent suspensions with and without dyes could be measured and used to elucidate better the influence of pH on the different adsorption rates for the tested materials;
Please, specify the uncertainty of the measurements in adsorption tests.
Author Response
Dear reviewer,
Subject: Submission of revised paper “polymers-1378669”
We wish to express our strong appreciation to you for careful reading our manuscript and for giving useful comments. We have carefully reviewed the comments and have revised the manuscript accordingly. The following is a point-by-point response to the comments.
Comments
- Written elements in Figure 3 are unreadable. Please, correct that.
Answer
Figure 3 was not successfully converted to a PDF file. We replaced it with a clearer figure. Similarly, Figures 7 to 10 have been replaced with clearer figures.
- Experiments of dye adsorption at different pH values should be better addressed. They are not refereed in the experimental part, but appear in Figure 11 (discussion) only for Congo red and in a short range of pH (3-5). Did the authors try to vary this parameter for other dyes and different pH ranges?
Answer
In reality, we examined the dye concentration reduction rate in the pH range of 3–12. Thus we added a graph of the dye concentration reduction rate at pH 8 in addition to Figure 11(b) that expresses the change in pH of the aqueous solution. Congo-red solution of pH 12 was adjusted by adding sodium hydroxide. The dye concentration reduction rate was almost comparable to that at pH 8. This experimental result was not shown in this manuscript.
We also examined the concentration reduction rates of MB and Orange II dyes when the pH of their dye solutions was adjusted in the range of 2 to 12. For MB dye, the reduction rate increased at pH 12. For Orange II dye, the reduction rate increased a little at pH 2. The result was almost as expected. The above results were also not shown in this manuscript because the adsorption tests on all samples have not yet been completed. We would like to report it in a later paper.
- In the same line with the last comment, zeta potential measurements on adsorbent suspensions with and without dyes could be measured and used to elucidate better the influence of pH on the different adsorption rates for the tested materials;
Answer
Thank you for your valuable comments. I would like to use the method in future research.
- Please, specify the uncertainty of the measurements in adsorption tests.
Answer
We added the following sentences in accordance with your point. Page 12, 341 line;
However, in this study, the dye concentration of the aqueous solution was indirectly estimated by measuring its absorbance. The color of Congo-red solution changes from red to dark blue when the pH is approximately 3 or less [28]. Namely, the color of the aqueous solution becomes darker. Therefore, when the pH of the aqueous solution is adjusted to 3 or less, the relationship between the absorbance and the dye concentration should be recalibrated, but this is not easy and is under consideration. For this reason, the dye concentrations reduction rate at pH 3 in Figure 11(a) may contain relatively large measurement errors. Moreover, the effect of carbides in the reductively fired GFRP/clay ceramic on the adsorption of Congo-red dye is still unknown. These issues should be addressed in the future.
- Others
In Figure 8 (In the upper MB diagram); we corrected “clay (oxidatively fired)” to “clay”.
Page 12, 337 line; we added “initial”.
Page 12, 338 line; we corrected “in the pH range of 3–5” to “in the range of 3–8”.
Page 12, 338 line and caption in Figure 11; we corrected “conductively” to “reductively”.
In addition, the characters written in red have been corrected.
Thank you once again for your valuable comments and suggestions. We are grateful for the time and energy you expended on our behalf.
Sincerely yours,
Hiroyuki Kinoshita
University of Miyazaki Japan

Reviewer 2 Report
The work aims at an important issue of finding new versatile agents for water purification techniques, namely reused waste glass fiber-reinforced plastic, produced by reductive or oxidizing firing of GFRP-clay mixture, and the adsorption mechanisms of dyes on their surface. There are several interesting findings, suggesting that the adsorption mechanism is based on ion exchange with the calcium component of the glass fibers, importance of specific surface area of the adsorbent and the effect of the plastic carbide residue upon it, adjusting the clay type for the specific dye, etc.
The strength of the work: Combination of the state-of the art sample preparation techniques with appropriate set of complimentary characterization tools. Clear and unambiguous interpretation and analysis of the data. Quite immediate application relevance of the results.
The weakness: The work would gain if the results were added by molecular level information relating to characterization of the adsorbents and the effect of dye adsorption on its surface, obtained, e.g., by FTIR.
In general, the work is scientifically sound, is logically and clearly conducted and presented, figures are informative and clear, reference list is appropriate and up-to-dated. In my view, the manuscript is suitable for publication in Polymers in its present form.
Author Response
Dear reviewer,
Subject: Submission of revised paper “polymers-1378669”
We wish to express our strong appreciation to you for careful reading our manuscript and for giving useful comments.
Comments
The work aims at an important issue of finding new versatile agents for water purification techniques, namely reused waste glass fiber-reinforced plastic, produced by reductive or oxidizing firing of GFRP-clay mixture, and the adsorption mechanisms of dyes on their surface. There are several interesting findings, suggesting that the adsorption mechanism is based on ion exchange with the calcium component of the glass fibers, importance of specific surface area of the adsorbent and the effect of the plastic carbide residue upon it, adjusting the clay type for the specific dye, etc.
The strength of the work: Combination of the state-of the art sample preparation techniques with appropriate set of complimentary characterization tools. Clear and unambiguous interpretation and analysis of the data. Quite immediate application relevance of the results.
The weakness: The work would gain if the results were added by molecular level information relating to characterization of the adsorbents and the effect of dye adsorption on its surface, obtained, e.g., by FTIR.
In general, the work is scientifically sound, is logically and clearly conducted and presented, figures are informative and clear, reference list is appropriate and up-to-dated. In my view, the manuscript is suitable for publication in Polymers in its present form.
Answer
- We wish to acknowledge your highly valuable comments on the necessity of molecular level analysis and FTIR usefulness.
Others
- In Figure 8 (In the upper MB diagram); we corrected “clay (oxidatively fired)” to “clay”.
- Page 12, 337 line; we added “initial”.
- Page 12, 338 line; we corrected “in the pH range of 3–5” to “in the range of 3–8”.
- Page 12, 338 line and caption in Figure 11; we corrected “conductively” to “reductively”.
- In addition, the characters written in red have been corrected.
- In Fig. 11, we added the dye concentration reduction rate at pH 8 and Figure 11(b) that expresses the change in the pH value of the aqueous solution in accordance with the other reviewer point.
Thank you once again for your valuable comments and suggestions. We are grateful for the time and energy you expended on our behalf.
Sincerely yours,
Hiroyuki Kinoshita
University of Miyazaki Japan
